# Ultrasensitive and Specific Detection of Anticancer Drug 5-Fluorouracil in Blood Samples by a Surface-Enhanced Raman Scattering (SERS)-Based Lateral Flow Immunochromatographic Assay

**DOI:** 10.3390/molecules27134019

**Published:** 2022-06-22

**Authors:** Hanwen Liu, Ying Liu, Ting Zhou, Penghui Zhou, Jianguo Li, Anping Deng

**Affiliations:** The Key Lab of Health Chemistry and Molecular Diagnosis of Suzhou, College of Chemistry, Chemical Engineering and Materials Science, Soochow University, Renai Road 199, Suzhou 215123, China; 20194209146@stu.suda.edu.cn (H.L.); ly3267047748@163.com (Y.L.); 20194209078@stu.suda.edu.cn (T.Z.); 20194209007@stu.suda.edu.cn (P.Z.)

**Keywords:** surface-enhanced Raman scattering (SERS), lateral flow immunochromatographic assay (LFIA), 5-fluorouracil (5-FU), anticancer drug, blood samples

## Abstract

5-Fluorouracil (5-FU) is an effective anticancer drug widely used in the world. To improve therapy efficiency and reduce side effects, it is very important to frequently detect the concentration of 5-FU in blood samples of patients. In this work, a new type of lateral flow immunochromatographic assay (LFIA) based on surface-enhanced Raman scattering (SERS) for ultrasensitive and specific detection of 5-FU in blood samples was developed. Au@Ag/Au nanoparticles (NPs) employing Au particles as the core and Ag/Au alloy as the shell were synthesized, characterized and used as the substrate in SERS-LFIA due to their high SERS enhancement and biocompatibility. The immunoprobe was made in the form of Au^MBA^@Ag/Au-Ab in which mercaptobenzoic acid (MBA, a common Raman active reporter) was embedded in the core–shell layer and the monoclonal antibody (mAb) against 5-FU was immobilized on the surface. The performance of SERS-LFIA was similar to that in colloidal gold based-LFIA, and the entire assay time was within 20 min. According to the color intensity on the testing (T) lines of LFIA strips visualized by eyes, the contents of 5-FU in the samples could be qualitatively or semi-quantitatively identified. Furthermore, by measuring the characteristic Raman intensities of MBA on T lines, quantitative detection of 5-FU in the samples were achieved. The IC_50_ and limit of detection (LOD) of the LFIA for 5-FU were found to be 20.9 pg mL^−1^ and 4.4 pg mL^−1^, respectively. There was no cross-reactivity (CR) of the LFIA with nine relative compounds, and the CR with cytosine, tegafur and carmofur were less than 4.5%. The recoveries of 5-FU from spiked blood samples were in the range of 78.6~86.4% with the relative standard deviation (RSD) of 2.69~4.42%. Five blood samples containing 5-FU collected from the Cancer Hospital were measured by SERS-LFIA, and the results were confirmed by LC-MS/MS. It was proven that the proposed method was able to simply and rapidly detect 5-FU in blood samples with high sensitivity, specificity, accuracy and precision.

## 1. Introduction

Pyrimidine compounds, including uracil, cytosine and thymine, are one kind of prominent nitrogen-containing heterocyclic substances existing in DNA or RNA chains in human being. In addition, the pyrimidine derivatives have been used as important drugs to treat many diseases because they exhibit a variety of biological activities such as antibacterial, anti-inflammatory, analgesic, antipyretic, anti-viral, anti-allergic and antioxidant [1,2]. In the last five years, 5-fluorouracil (5-FU, a derivative of uracil) has been widely used in the treatment of many solid tumors such as head, neck and gastrointestinal tract tumors [3,4]. However, the overdose of 5-FU in blood can cause serious side effects such as chest pain, arrhythmia, myocarditis and pericarditis, heart failure and even death [5,6]. On the other hand, a low dose of 5-FU in blood will the decrease efficiency of therapy. Therefore, it is urgently required to develop a highly sensitive and specific analytical method for detection of 5-FU in blood samples in simple, rapid and real-time performance.

Although many analytical methods for the detection of 5-FU in blood samples have been reported, high performance liquid chromatographic (HPLC) methods with different detectors, such as MS, LC-MS/MS, UV and DAD, are the most common analytical methods for 5-FU detection in clinical applications [7,8,9,10]. The chromatographic methods are accurate, but they are expensive, have a high testing cost, low throughput and are time-consuming due to requiring a complicated sample pretreatment before analysis.

Surface-enhanced Raman scattering (SERS) is a phenomenon where Raman active molecules are adsorbed on the surface of a rough noble metals (gold, silver, etc.), the Raman intensities of the molecules absorbed on metals can be greatly amplified [11]. There are two main explanations for the enhancement mechanism of SERS currently accepted, namely electromagnetic (EM) and charge transfer (CT) theories. The EM theory considers that the enhancement mainly results from the interactions between the molecules and the huge surface resonance plasmons on the surface of the substrate induced by incident laser [12]. The CT theory claims the enhancement also involves a strong charge transfer between the molecules and the substrate in the electronic resonance transition [13]. The high enhancement efficiency makes the SERS the most sensitive analytical technique even for the detection of a single molecule [14,15,16].

Synthesis of noble metal substrates with high enhancement efficiency is the key work to develop a SERS-based analytical method. Most of the substrates are made from Au NPs or Ag NPs [17,18]. Au NPs can be simply prepared with good biocompatibility and middle SERS enhancement. Although Ag NPs display high SERS enhancement, their biocompatibility and stability are poor. The SERS enhancement effect can be attributed to the surface tip structure called “hot spots”. Currently many materials made from noble metals with sufficient “hot spots” in the form of rods, flowers, prisms, cubes, cones, etc. have been produced [19]. The shell gap between the spheres can also be considered as “hot spots”. In addition, the core–shell bimetallic nanoparticles were found to have a higher SERS activity due to electronic ligand effect and localized electric field enhancement in core–shell NPs [20].

Lateral flow immunochromatographic assay (LFIA) has been by far the most commonly used in point-of-care testing (POCT) setting due to the affordability, effectiveness and short-time detection [21]. Usually in the colloidal gold-based LFIA, the immunoprobe is simply made by immobilizing antibody (Ab) on the surface of colloidal gold. Due to colloidal gold showing a bright red color, the color intensity on T lines observed by human eyes can only be used for qualitative or semi-quantitative detection. In the last decade, a quantitative LFIA using its fluorescence as a signal readout based on fluorescent NPs of microspheres, quantum dots and lanthanide NPs has received great attention [22,23,24], but fluorescent-based LFIA exhibits some shortcomings such as complicated preparation of NPs, instability of fluorescence and inferences from biological substances in blood samples. Recently, a novel ultrasensitive and quantitative LFIA using SERS as signal readout has been developed with great progress. SERS-LFIA has been widely used in clinical, biological, food and environmental analytical fields [25,26,27,28,29,30,31,32,33]. In addition, the portable SERS-LFIA readers have been designed and manufactured [34]. Currently, the emphasis of SERS-LFIA research is mainly on the synthesis of noble metal substrates with high enhancement used as the substrates for the preparation of immunoprobes.

In this study, Au@Ag/Au NPs employing Au particles as the core and Ag/Au alloy as the shell were synthesized, characterized and used as the substrate due to their high SERS enhancement and biocompatibility. The immunoprobe was prepared by immobilizing mAb against 5-FU on the surface of Au@Ag/Au NPs, and embedding the Raman reporter MBA between the core/shell in order to protect the loss of the reporter. After the LFIA procedures, qualitative or semi-quantitative identification of 5-FU was made by observing the color intensity on T lines of LFIA strips. More importantly, by measuring the Raman intensities of MBA on T lines, ultrasensitive and quantitative detection of 5-FU was achieved. The SERS-LFIA was also applied for the detection of 5-FU in blood samples collected from the Cancer Hospital, and the results were confirmed by HPLC-MS/MS.

## 2. Materials and Methods

### 2.1. Chemicals, Materials and Apparatus

Hexadecyltrimethylammonium chloride (CTAC) was purchased from TCI Co., Ltd. (Shanghai, China). Silver nitrate (AgNO_3_, 99.8%) was obtained from the Sinopharm Chemical Reagent Co., Ltd. (Shanghai, China). L-Ascorbic acid (AA, 99%), casein, 5-FU and twelve pyrimidine compounds (cytosine, tegafur, carmofur, thymine, capecitabine, 5-fluoro-2’-deoxyuridine, uracil, 5-bromouracil, 5-fluoro-1,3-dimethyluracil, uridine, 5-bromo-2’-deoxyuridine and gimeracil) for testing cross-reactivity (CR) were purchased from Aladdin (Shanghai, China). Sodium citrate tribasic dihydrate (Na_3_C_6_H_5_O_7_·2H_2_O, 99.5%), chloroauric acid (HAuCl_4_), 4-mercaptobenzoic acid (4-MBA) and ovalbumin (OVA) were purchased from Sigma (St. Louis, MO, USA). All other chemicals were analytical grade. The mAb against 5-FU and coating antigen (5-FU-OVA) were prepared by our group.

Nitrocellulose (NC) membranes were purchased from Whatman (Shanghai, China). PVC sheets, filter paper and adhesive tape were obtained from Jieyi Biotechnology Co. Ltd. (Shanghai, China). The deionized-RO water supply system (Dura 12FV) was purchased from THE LAB Com. (Dover, DE, USA). The UV-2300 spectrophotometer was purchased from Techcom (Shanghai, China). The digital photographs of the samples were taken with Mi 8 (Suzhou, China). Transmission electron microscopy (TEM) photographs were taken on a Tecnai G220 from USA FEI Company. The portable Raman Analyzer RamTracer-200-HS was obtained from Opto Trace Technologies, Inc. (Suzhou, China). HPLC-MS/MS was performed with Shimadzu HPLC system consisting of a DGU-20A3 degasser, two LC-20AD pumps, a SIL-20ACHT autosampler and a CTO-20AC column temperature oven (Shimadzu corporation, Kyoto, Japan).

### 2.2. Synthesis of Au^MBA^@Ag/Au NPs

All of the glassware used in this experiment was cleaned with freshly prepared aqua regia (HNO_3_/HCl, 1:3, *v*/*v*), then thoroughly rinsed ten times with tap water and ultrapure water.

The synthesis of Au^MBA^@Ag/Au NPs was illustrated in Figure 1a. Firstly, the spheroidal Au NPs were prepared according to the literature with a slight modification [35]. Briefly, 100 mL of ultrapure water was heated to boiling followed by dispersing 0.2 mL of 5% (w.t.) HAuCl_4_ under a vigorous stirring. Then 1.5 mL of freshly prepared Na_3_C_6_H_5_O_7_·2H_2_O was rapidly added into the solution which was boiled again. To ensure the reduction completed, the solution was kept in boiling for 10 min until no color change, then cooled down to room temperature. Secondly, 200 µL of MBA (0.1 mmol L^−1^) was added to the solution containing Au NPs and the mixture was under stirring for 3 h at room temperature so that MBA was tightly adsorbed onto the surface of Au NPs via thiol group. The formed Au^MBA^ NPs was centrifuged at 8000 rpm for 10 min to remove any excess reagents. The precipitate was redispersed in 80 mL of ultrapure water and stored at 4 °C until use. Thirdly, 3 mL of AA (0.1 mol L^−1^), 2 mL of CTAC (0.2 mol L^−1^), 5 mL of H_2_O and 2 mL of AgNO_3_ (0.01 mol L^−1^) were added to 20 mL of Au^MBA^ NPs solution. The mixture was kept at 60 °C for 15 min with slight stirring to form Au^MBA^@Ag NPs. Finally, 0.1 mL of HAuCl_4_ (0.05 mol L^−1^) was added to Au^MBA^@Ag NPs solution, and the mixture was kept at 60 °C for another 15 min with slight stirring to form Au^MBA^@Ag/Au NPs. During this process, HAuCl_4_ etched the Ag shell and excessive AA reduced HAuCl_4_ to form Ag/Au alloy shell around Au^MBA^ NPs. The final solution was centrifuged at 7000 rpm for 10 min to eliminate any excess reagents. The sediment was washed with H_2_O twice and redispersed in ultrapure water. The Au^MBA^@Ag/Au NPs solution was stored at 4 °C until use.

### 2.3. Preparation of Immunoprobe

As shown in Figure 1a, in the processes of immunoprobe (Au^MBA^@Ag/Au-Ab) preparation, Au NPs were firstly synthesized as the core, then MBA was adsorbed directly onto the surface of Au NPs to form Au^MBA^ NPs. In presence of Au^MBA^ NPs, by reducing AgNO_3_ and HAuCl_4_ with ascorbic acid, Ag/Au alloy was formed which was as the shell around the Au^MBA^ NPs. Then 8 μL of mAb against 5-FU (1.12 mg mL^−1^) was added to 4 mL of Au^MBA^@Ag/Au NPs solution with gently stirring at 4 °C for 4 h. Finally, 50 μL of 5% casein was added to the mixture and stirring for 1.5 h to block the unspecific binding sites on the surface of the NPs. The mixture solution was centrifuged at 3000 rpm for 15 min at 4 °C, and the sediment was redispersed in an equal volume of water. The final immunoprobe Au^MBA^@Ag/Au-Ab was stored at 4 °C until use.

### 2.4. Fabrication of LFIA Strip

As illustrated in Figure 1b, the LFIA strip consisted of five parts including a sample pad for sample addition, a conjugate pad containing Au^MBA^@Ag/Au-Ab, an absorbent pad, a NC membrane with a T line and a C line, and a PVC bottom plate. The absorbent pad is 100% pure cellulose fiber which can provide high absorbent capacity. The T line and C line on NC membrane were formed by evenly distributing 4 μL of coating antigen (5-FU-OVA, 1 mg mL^−1^) and 4 μL of the second antibody (goat anti-mouse IgG, 1:30 dilution), respectively. The NC membrane was fixed on the middle of the PVC bottom plate. The sample pad and the absorbent pad were pasted at both ends of the NC membrane with an overlap of about 1–2 mm. After naturally drying at room temperature for 1 h, the LFIA strips were sealed into glass bottles in the presence of nitrogen gas and desiccant gel and stored at 4 °C until use.

### 2.5. Procedures of Competitive LFIA

As shown in Figure 1b, when 200 μL of 5-FU standard or sample solution is dropped onto the sample pad, the solution will migrate toward absorbent pad because of capillary action. The solution will be firstly combined with Au^MBA^@Ag/Au-Ab on the conjugate pad, then flows through the T line and C line, and finally reaches the absorbent pad. The whole LFIA procedures can be completed in 20 min. In negative situation (no 5-FU), the Au^MBA^@Ag/Au-Ab on conjugate pad will move to T line and specifically bind with coating antigen (5-FU-OVA) because of specific reaction between antibody and antigen. The superfluous immunoprobe will continuously flow to C line and be captured by the second antibody. Under this circumstances, two dark-red bands on T line and C line will be appeared. On the contrary, in positive situation (the concentration of 5-FU is high), during the migration, the Au^MBA^@Ag/Au-Ab on conjugate pad will be firstly bound with high concentration of 5-FU, leaving no free antibody binding sites for coating antigen on T line. Therefore, there was almost no Au^MBA^@Ag/Au-Ab on the T line. However, Au^MBA^@Ag/Au-Ab-5-FU can continue flow to C line and be captured by the second antibody, showing a dark-red color. Apparently, the higher concentration of 5-FU, the lesser of Au^MBA^@Ag/Au-Ab bound on T line. In other words, the degree of intensity of immunoprobe color on the T line is inversely proportional to the concentration of 5-FU. According to the color intensity on T lines visualized by eyes, the contents of 5-FU in the samples can be qualitatively or semi-quantitatively identified. Furthermore, quantitative detection of 5-FU in the samples can be achieved by measuring the characteristic Raman intensities of MBA at 1588 cm^−1^ on T lines. A portable Raman Analyzer coupled with a microscope (Eplan, 40 × 0.6), an excitation source at 785 nm with a laser power of 50 mW and a typical integration time of 10 s was used for Raman signals. Average SERS intensity from 10 different spots along the T line was collected for quantification.

### 2.6. Detection of 5-FU in Spiked Samples

To examine the accuracy and precision of the LFIA, the blank human blood samples were used for spiking experiment. After centrifuging the blood samples, the supernatants, e.g., sera, were collected. To four tubes containing 200 μL of serum, different amount of 5-FU solution was added. The tubes were vortexed for 30 s, and then 1 mL ethyl acetate was added. After being vortexed for 3 min again, the tubes were centrifuged at 12,000 rpm for 7 min. The supernatants were removed to other four tubes and evaporated to dryness in a 45 °C water bath under nitrogen. The residues were redissolved in 600 μL of 0.01 mol L^−1^ phosphate buffered saline (PBS, pH = 7.4) to ensure the final concentrations of 5-FU were 0, 0.01, 0.1 and 1 ng mL^−1^, respectively. Then 200 μL of the diluted sample was applied to LFIA procedures. The recovery of 5-FU from the spiked samples was calculated as: Recovery (%) = [(5-FU concentration measured − Blank)/5-FU concentration spiked] × 100%.

### 2.7. LC-MS/MS Analysis

LC-MS/MS was performed as follows. The mobile phase consisted of methanol and water (2:98, *v*/*v*) with isocratic elution, and the flow rate was 0.30 mL min^−^^1^. Separation was carried out on an Agela Innoval NH_2_ column (2.1 mm × 50 mm, 5 μm) maintained at 40 °C. The injection volume was 10 μL. Mass spectrometric detection was conducted on an AB SCIEX 3200 system (Applied Biosystem Analytical Technologies, Foster City, CA, USA) equipped with an electrospray ionization (ESI) source. The mass spectrometer was operated in negative mode and 5-bromouracil (5-BR) was used as an internal standard. Quantification was performed by multiple reaction monitoring (MRM). The LC-MS/MS setting parameters were as follows: −28 eV (5-FU) and −31 eV (5-BR) ion source collision energy, −26 V (5-FU) and −28 V (5-BR) declustering potential (DP). The data acquisition and processing were performed with Analysts 1.5 software (Applied Biosystems Analytical Technologies, Foster City, CA, USA). The selected MRM transitions for 5-FU were *m*/*z* 128.8/42.1 (5-FU) and 186.2/−42.1 (5-BR) with a dwell time of 100 ms. The calibration curve for 5-FU was constructed with standards of 10, 20, 50, 100, 200, 500 and 1000 ng mL^−1^.

## 3. Results and Discussion

### 3.1. Synthesis and Characterization of Different NPs

Usually, Au NPs can be easily obtained according to normal procedures. MBA can be attached to Au NPs via the thiol group, thus the preparation of Au^MBA^ NPs is not difficult. However, excess addition of MBA will cause Au^MBA^ NPs to be precipitated. In this study, 200 µL of MBA at the concentration of 0.1 mmol L^−1^ is considered the best amount to be added to 10 mL of Au NPs. In the preparation of Au^MBA^@Ag/Au NPs, when different volumes of AgNO_3_ solution (100 µL, 150 µL, 200 µL, 250 µL) and the same volume of HAuCl_4_ solution were added to four glass bottles containing 2 mL of Au^MBA^ NPs, four kinds of Au^MBA^@Ag/Au NPs with different size were obtained.

The TEM images and UV-vis spectra of the Au^MBA^ NPs, Au^MBA^@Ag NPs and Au^MBA^@Ag/Au NPs with different sizes are illustrated in Figure 2 and Figure 3a, respectively. It is seen from Figure 2a that the Au^MBA^ NPs are homogeneous with a size of about 25 nm. As shown in Figure 3a (black line), the Au^MBA^ NPs displays an absorption peak at 528 nm, which is typical evidence of Au NPs. It can be seen from Figure 2b that the Au^MBA^@Ag NPs are also uniform with a size of about 64 nm. Thus, the thickness of the Ag shell in Au^MBA^@Ag NPs is calculated to be (64 − 25)/2 = 19.5 nm. While from Figure 3a (red line), there are two absorption peaks at 405 nm and 510 nm, which clearly demonstrates that Au NPs were successfully covered by Ag shell because the absorption peak of Ag NPs normally appears at 400 nm. As shown in Figure 2c–f, the sizes of the four Au^MBA^@Ag/Au NPs are found to be 36, 44, 48 and 52 nm, respectively. Therefore, the thicknesses of the Ag/Au alloy shell on these NPs are estimated to be 5.5 nm, 9.5 nm, 11.5 nm and 13.5 nm, respectively. The UV-vis spectra of Au^MBA^@Ag/Au NPs with the sizes of 5.5 nm, 9.5 nm, 11.5 nm and 13.5 nm are shown in Figure 3a with blue, green, pink and yellow lines, respectively. It can be seen that with the thickness of Ag/Au alloy shell increasing (from 5.5 nm to 13.5 nm), the absorption peaks of Au^MBA^@Ag/Au NPs exhibit a red-shift from 406 nm to 457 nm, which coincide well with the traditional Mie scattering theory and dielectric data [36].

The Raman intensities of Au^MBA^ NPs, Au^MBA^@Ag NPs and Au^MBA^@Ag/Au NPs (size 44 nm) are illustrated in Figure 3b. It is obvious that among three NPs, Au^MBA^@Ag/Au NPs display the highest enhancement effect, mainly due to the gap of the electromagnetic field between Ag/Au alloy shell and Au. Meanwhile, the Raman intensities of four kinds of Au^MBA^@Ag/Au NPs are shown in Figure 3c. It is seen from Figure 3c that with the Ag/Au shell thickness increasing, the Raman intensities are increased, and arrived at the highest signals point with the Ag/Au alloy shell thickness of 9.5 nm. Further increasing shell thickness results in rapidly signal decline, mainly due to the thicker Ag/Au shell hindering the output of the Raman signal. Thus, the Au^MBA^@Ag/Au NPs (size 44 nm) was chosen and used in SERS-LFIA.

### 3.2. Characterization of Immunoprobe

Before the detection of 5-FU, the specific binding of immunoprobe with the coating antigen (5-FU-OVA) on the LFIA strip should be tested. Four LFIA procedures were performed at zero concentration of analyte under following four situations: (a) Au^MBA^@Ag/Au-Ab as probe, 5-FU-OVA dispersed on the T line; (b) Au^MBA^@Ag/Au-casein as probe, 5-FU-OVA dispersed on the T line; (c) Au^MBA^@Ag/Au-Ab as probe, OVA coated on the T line; (d) Au^MBA^@Ag/Au-Ab as probe, Na_2_CO_3_-NaHCO_3_ buffer solution coated on the T line. The Raman spectra in the above four cases are illustrated in Figure 4. From spectrum (a) in Figure 4, it can be seen that the specific Raman scattering peak of the MBA at 1588 cm^−1^ with the signal values of 12,809 (a.u.) appears, which clearly demonstrates that the immunoprobe can be specifically captured by 5-FU-OVA at the T line on the NC membrane. In contrast, from the Raman spectra (b–d) in Figure 4, almost no SERS signals appear at 1588 cm^−1^. The lack of SERS intensities from spectrum (b) is due to the lack of antibody on the probe, while lack of SERS intensities from spectrum (b) and (c) indicates that when the T line is coated with OVA (or Na_2_CO_3_-NaHCO_3_ buffer) instead of coating the antigen, the immunoprobe is not able to be appeared on the T line.

### 3.3. Optimization of Experimental Conditions

The sensitivity of the SERS-LFIA can be improved by optimizing experimental parameters such as the amount of Ab used for the immunoprobe prepared, the amount of the coating antigen and the dosage of immunoprobe applied on the strip. In this study, the inhibition ratio (B_0_/B_0.1_) is defined and used to evaluate the optimization, where B_0_ and B_0.1_ refer to the SERS intensities of MBA at 1588 cm^−1^ when 5-FU concentrations are at 0 ng mL^−1^ and 0.1 ng mL^−1^, respectively. Higher value of B_0_/B_0.1_ reveals the higher sensitivity of the assay.

During immunoprobe preparation, 0.5~3.0 µL of Ab against 5-FU (1.12 mg mL^−1^) were consecutively coupled to 1.0 mL of Au^MBA^@Ag/Au NPs. The effect of the amount of Ab on the inhibition ratio B_0_/B_0.1_ was investigated. It is observed from Figure 5a that the highest value of the B_0_/B_0.1_ is achieved at 1.5 μL. Thus, the optimal amount of Ab for immunoprobe preparation was 1.5 μL at the concentration of 1.12 mg mL^−1^.

The amount of the 5-FU-OVA coated on the T line and the immunoprobe applied on the conjugate pad also have an important effect on the assay. When the volume of 5-FU-OVA was set at 4.0 μL, the effect of the concentration of 5-FU-OVA in the range of 0.1–2.5 mg mL^−1^ on the assay was examined. As shown in Figure 5b, the highest value of the inhibition ratio B_0_/B_0.1_ is achieved at 1.0 mg mL^−1^. Similarly, it is observed from Figure 5c that when 1.5~4.0 μL of immunoprobe is dropped on the conjugate pad, the highest value of the B_0_/B_0.1_ is achieved at 2.5 μL. Thus, the optimal conditions for SERS-LFIA are: 4 μL of 5-FU-OVA (1.0 mg mL^−1^) coated on the T line and 2.5 μL of immunoprobe applied on the conjugate pad.

### 3.4. Sensitivity of SERS-LFIA

5-FU standard solutions (0, 10^−4^, 10^−3^, 10^−2^, 10^−1^, 1.0, 10 and 100 ng mL^−1^) were prepared by diluting 5-FU stock solution (1 mg mL^−1^) with PBS and applied to SERS-LFIA procedures under optimal assay conditions. Two hundred microliters of 5-FU standards was added to sample pad which would move toward the end of absorption pad. At about 20 min, as shown in Figure 6a, the colors on T line and C line gradually appeared and were seen directly by the naked eye. The color intensity visualized by eye at the T line is inversely related to the concentration of 5-FU solution, which can be used for qualitative or semi-quantitative identification. For quantitative detection, the SERS spectra of MBA from immunoprobe captured on the T lines were measured by SERS analyzer. As shown in Figure 6b, the SERS intensities at 1588 cm^−1^ generated from MBA were gradually declined with the increasing of 5-FU concentration. The standard curve of the SERS-LFIA for 5-FU was plotted in form of B/B_0_ × 100% versus log C, in which B and B_0_ were the SERS intensity of MBA at the standard point and zero concentration, respectively (Figure 6c). The sensitivity expressed by IC_50_ value, e.g., the concentration of 5-FU producing 50% signal inhibition, was calculated to be 20.9 pg mL^−1^. The LOD at three times of standard deviation (SD) was estimated to be 4.4 pg mL^−1^. IC_50_ and LOD values are very low, indicating high sensitivity of the SERS-LFIA for 5-FU. In cancer treatment, the effective dose of 5-FU in blood samples is about 150 ng mL^−1^ [37]. Apparently, the sensitivity of SERS-LFIA is sufficiently satisfied with the requirement for 5-FU detection.

In the last decade, many analytical techniques have been developed for the detection of 5-FU. The comparison of different analytical techniques including our SERS-LFIA for the detection of 5-FU is presented in Table 1. It can been seen from Table 1 that the LOD value achieved in our work is much lower than that in other analytical techniques, indicating the ultrasensitivity of our assay.

### 3.5. Reproducibility of SERS Intensities

The reproducibility of the SERS intensities was researched by comparing the signals at 1588 cm^−1^ generated from MBA measured from ten different spots in the center sections of the T line when 5-FU standard solutions were at 0, 0.1 and 1.0 ng mL^−1^, respectively. As illustrated in Figure 7, it is apparent that the higher concentration of 5-FU will lead to the lower SERS intensities. In each strip, the RSD values of the SERS intensities obtained from 10 different points were 5.22%, 6.93% and 6.30%, respectively, suggesting high precision of the SERS signal.

### 3.6. Specificity of SERS-LFIA

The specificity of a competitive immunoassay was often characterized by the cross-reactivity (CR) value. Twelve pyrimidine compounds were selected to evaluate the specificity of the SERS-LFIA for 5-FU. All compounds including 5-FU were prepared in concentrations range of 10^−4^–10^3^ ng mL^−1^ and were subjected to SERS-LFIA procedures. The CR values were calculated using the equation of CR (%) = [IC_50_ of 5-FU)]/[IC_50_ of tested compound] × 100%. The obtained CR values were illustrated in Table 2. It was seen clearly that CR values of the assay with thymine, capecitabine, uracil, 5-fluoro-2’-deoxyuridine, 5-fluoro-1,3-dimethyluracil, uridine, 5-bromo-2’-deoxyuridine, 5-bromouracil and gimeracil were lesser than 0.01%, which is considered negligible. However, because of similarity in molecular structure, the assay displayed 4.40%, 2.96% and 4.00% of CR with cytosine, tegafur and carmofur, respectively.

### 3.7. Detection of 5-FU in Spiked Samples

In order to demonstrate the practicality of the SERS-LFIA, human serum was spiked with different concentration of 5-FU. After the pretreatment of the spiked sample, the obtained solutions were subjected to SERS-LFIA procedures. The results of the spiking experiment are summarized in Table 3. It was seen that the recoveries of the 5-FU from spiked samples are 78.6–86.4% with RSD in the range of 2.69–4.42% (*n* = 3), revealing that the proposed method was able to detect 5-FU in human serum samples with high accuracy and precision.

### 3.8. Real Samples Analysis

Five real blood samples were collected from patients in Cancer Hospital. After centrifugation to remove the precipitate, the sera were firstly pretreated, and then were detected by SERS-LFIA and LC-MS/MS simultaneously. The results measured by LC-MS/MS and SERS-LFIA are presented in Table 4. Clearly, there is a good correlation between two methods with the linear regression equation of Y = 0.7672X + 212.61 (R^2^ = 0.9851).

## 4. Conclusions

A quantitatively ultrasensitive and specific LFIA based on SERS for the detection of 5-FU in human blood samples was developed. The Au@Ag/Au core–shell bimetallic nanoparticles were synthesized, characterized and used as the substrate due to their high SERS enhancement and biocompatibility. Immunoprobe (Au^MBA^@Ag/Au-Ab) was prepared by embedding MBA in the core–shell layer and immobilizing mAb against 5-FU on the surface. The LFIA procedure was completed within 20 min. Color intensities observed by the naked eye on the T lines can be used for qualitative or semi-quantitative identification. Meanwhile, the specific SERS intensities of MBA at 1588 cm^−1^ on the T lines can be used for quantitative detection. Under optimal conditions, the IC_50_ and LOD values of the SERS-LFIA for 5-FU were 20.9 pg mL^−1^ and 4.4 pg mL^−1^, respectively, indicating the high-sensitivity of the assay. No CR of the LFIA with nine relative compounds, as well as less than 4.5% CR with cytosine, tegafur and carmofur, demonstrated the high specificity of the assay. The recoveries of the 5-FU in spiked samples were 78.6–86.4% with RSD valuea in the range of 2.69–4.42% (*n* = 3). Five blood samples collected from cancer patients were measured by SERS-LFIA, and the results were confirmed by LC-MS/MS. There was a good correlation between the two methods with the linear regression equation of Y= 0.7672X + 212.61 (R^2^ = 0.9851). It was proven that the proposed method was able to sensitively, specifically, simply and rapidly detect of 5-FU in clinical application. The platform can be also an alternative platform for the detection of other target analytes using corresponding Abs.

## Figures and Tables

**Figure 1 molecules-27-04019-f001:**
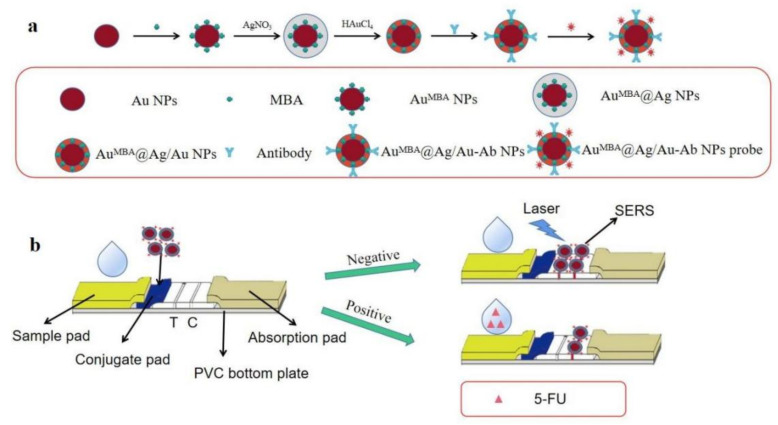
(**a**) Schematic illustration of the preparation of immunoprobe (Au^MBA^@Ag/Au-Ab); (**b**) Assembly of LFIA strip and the principle of competitive SERS-LFIA for 5-FU.

**Figure 2 molecules-27-04019-f002:**
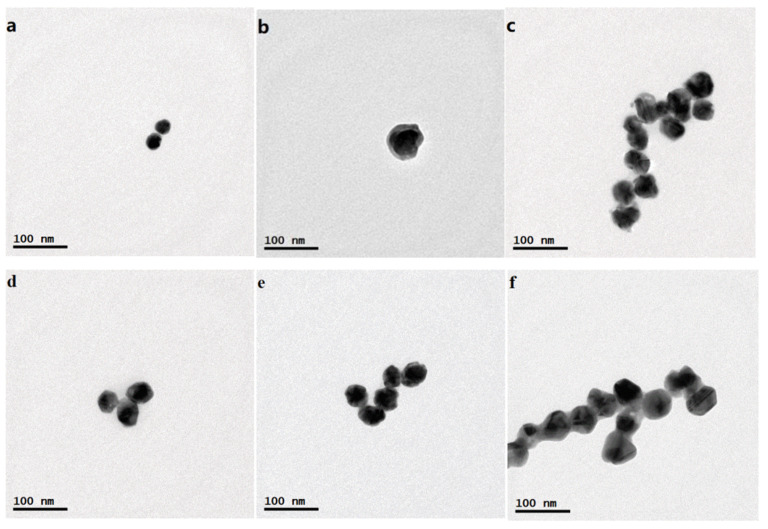
TEM images of different NPs: (**a**) Au^MBA^ NPs (25 nm); (**b**) Au^MBA^@Ag NPs (64 nm); (**c**–**f**) Au^MBA^@Ag/Au NPs with the size of 36 nm, 44 nm, 48 nm and 52 nm, respectively.

**Figure 3 molecules-27-04019-f003:**
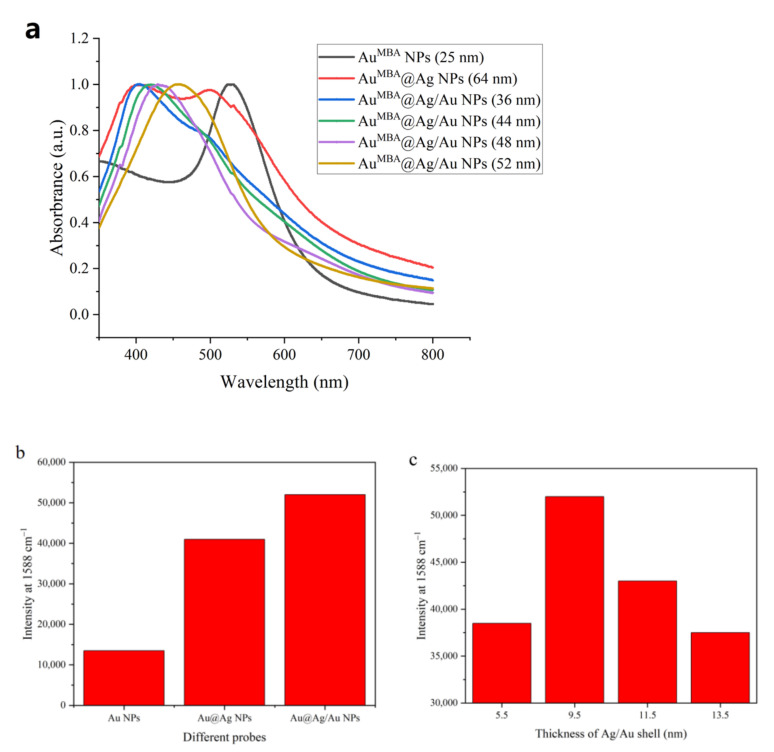
(**a**) UV-vis spectra of Au^MBA^ NPs (25 nm), Au^MBA^@Ag NPs (64 nm) and Au^MBA^@Ag/Au NPs with the size of 36 nm, 44 nm, 48 nm and 52 nm, respectively; (**b**) SERS intensities of Au^MBA^ NPs (25 nm), Au^MBA^@Ag NPs (64 nm) and Au^MBA^@Ag/Au NPs (44 nm); (**c**) Effects of the thicknesses of Ag/Au alloy shell in Au^MBA^@Ag/Au NPs on SERS intensities.

**Figure 4 molecules-27-04019-f004:**
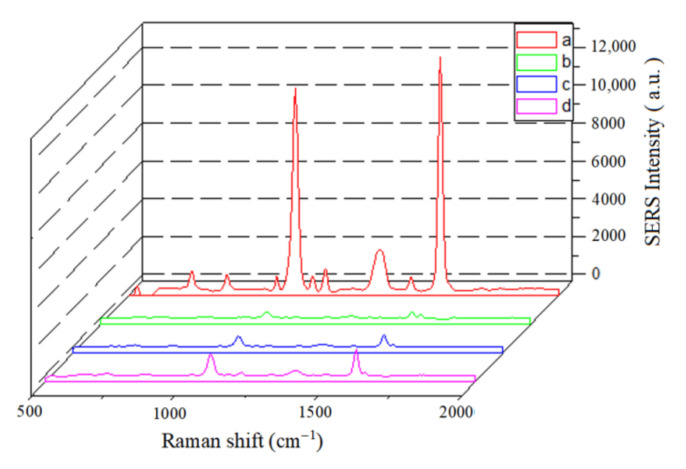
SERS spectra of SERS-LFIA on T line at 0 ng mL^−1^ of analyte based on different situations. (**a**) Au^MBA^@Ag/Au-Ab as probe, 5-FU-OVA dispersed on the T line; (**b**) Au^MBA^@Ag/Au-casein as probe, 5-FU-OVA dispersed on the T line; (**c**) Au^MBA^@Ag/Au-Ab as probe, OVA coated on the T line; (**d**) Au^MBA^@Ag/Au-Ab as probe, Na_2_CO_3_-NaHCO_3_ buffer solution coated on the T line.

**Figure 5 molecules-27-04019-f005:**
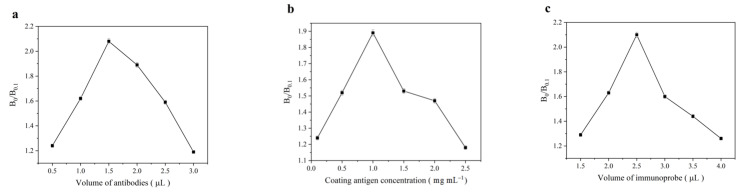
Optimization of the assay conditions: (**a**) Effect of antibody volume in the preparation of immunoprobe on B_0_/B_0.1_; (**b**) Effect of coating antigen on B_0_/B_0.1_; (**c**) Effect of the amount of immumoprobe on B_0_/B_0.1_.

**Figure 6 molecules-27-04019-f006:**
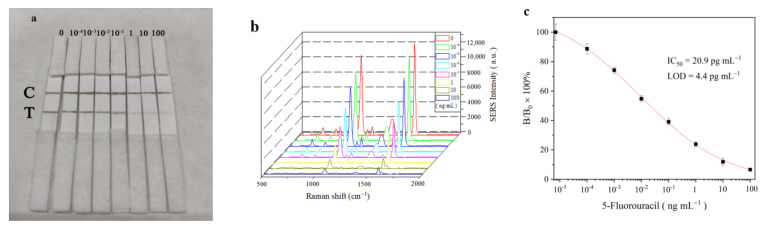
(**a**) Digital photograph of SERS-LFIA strips after the assay procedures. The numbers above the T lines are the standard concentrations of analyte (ng mL^−1^). (**b**) The SERS spectra arising from MBA on T lines after assay procedures at the standard concentrations. (**c**) The calibration curve of the SERS-LFIA for 5-FU, where B and B_0_ were the SERS intensities of MBA at 1588 cm^−1^ when the 5-FU solutions were at standard concentration and 0 ng mL^−1^, respectively.

**Figure 7 molecules-27-04019-f007:**
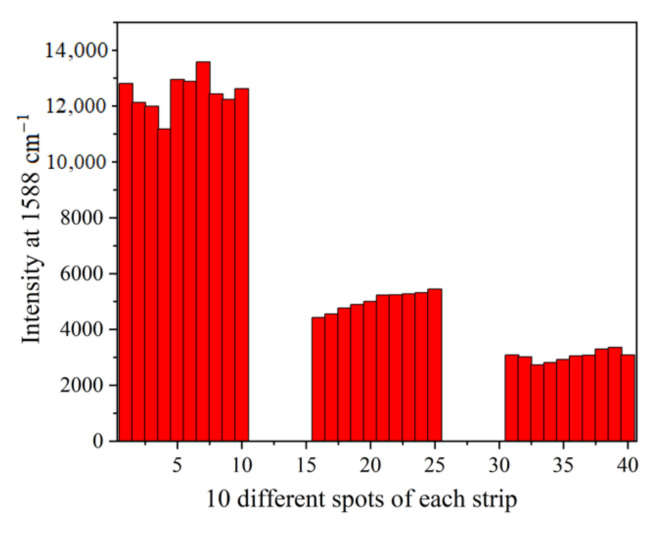
The SERS intensities of MBA at 1588 cm^−1^ from 10 different spots along the middle parts of the T lines on the strips when the concentrations were at 0, 0.1 and 1.0 ng mL^−1^. The RSD values of the SERS intensities were 5.22%, 6.93% and 6.30%, respectively.

**Table 1 molecules-27-04019-t001:** Comparison of different methods for 5-FU detection.

Detection Method	Linear Range (ng mL^−1^)	Detection Limit (ng mL^−1^)	Reference
Electrochemical sensor	13–64,000	3.9	[38]
Ratiometric PL sensor	0–130	2.6	[39]
Square wave voltammetry	1.3–6500	127.4	[40]
Ratiometric fluorescence detection	13–130,000	8.38	[41]
SERS	1300–130,000	1300	[42]
Electrochemical detection	65–15,600	2.47	[43]
Fluorescence detection	0–13,000	304.2	[44]
SERS-LFIA	0.0001–100	0.0044	This work

**Table 2 molecules-27-04019-t002:** The cross-reactivity (CR) values of the SERS-LFIA with 5-FU and other tested compounds.

Compound	Chemical Structure	IC_50_ (ng mL^−1^)	CR (%)
5-Fluorouracil	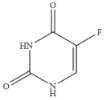	0.021	100
Cytosine	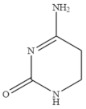	0.475	4.40
Tegafur	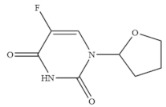	0.706	2.96
Carmofur	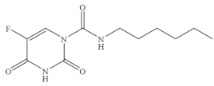	0.523	4.00
Thymine	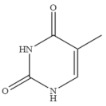	>100	<0.01
Capecitabine	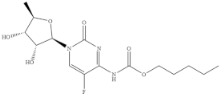	>100	<0.01
Uracil	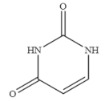	>100	<0.01
5-Bromouracil	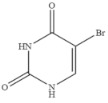	>100	<0.01
5-Fluoro-2’-deoxyuridine	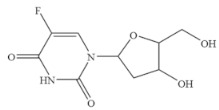	>100	<0.01
5-Fluoro-1,3-dimethyluracil	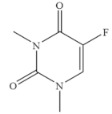	>100	<0.01
Uridine	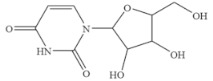	>100	<0.01
5-Bromo-2’-deoxyuridine	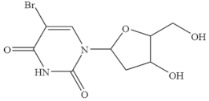	>1000	<0.001
Gimeracil	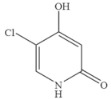	>1000	<0.001

**Table 3 molecules-27-04019-t003:** The recoveries of 5-FU from spiked blood samples measured by SERS-LFIA.

Conc. Spiked (ng mL^−1^)	Conc. Measured (ng mL^−1^) (Mean ± SD, *n* = 3)	RSD (%)	Recovery (%)
0.01	(0.786 ± 0.034) ×10^−2^	4.42	78.6
0.1	(0.863 ± 0.028) ×10^−1^	3.27	86.4
1	0.805 ± 0.021	2.69	80.6

**Table 4 molecules-27-04019-t004:** Comparison of LC-MS/MS and SERS-LFIA for the detection of 5-FU in blood samples collected from cancer patients.

No. of Sample	LC-MS/MS (ng/g)	SERS-LFIA	
		Mean ± SD (ng/g)	CV (%) (*n* = 3)
1	1160.5	1296.6 ± 98.5	7.6
2	315.7	344.5 ± 31.6	9.2
3	643.6	626.6 ± 42.6	6.8
4	3961.3	3210.7 ± 157.3	4.9
5	644.7	744.8 ± 55.9	7.5

## Data Availability

Not applicable.

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
