# Peer review of "Ultrasensitive and Specific Detection of Anticancer Drug 5-Fluorouracil in Blood Samples by a Surface-Enhanced Raman Scattering (SERS)-Based Lateral Flow Immunochromatographic Assay"

_molecules, 2022, doi:10.3390/molecules27134019_

Round 1

Reviewer 1 Report

The manuscript describes the development of a SERS lateral flow immunochromatographic assay for 5-FU detection. The content is scientifically sound and adequately presented. Nevertheless, I believe that the manuscript requires thorough revision, involving also language checking due to many instances of misspelling and truncated sentences. Moreover, I kindly ask authors to add references to support some statements as well as their observations. Some instances of points to be addressed are listed below:

-        Lines 37-38, the sentence: “heterocyclic substances existed in DNA or RNA chains in human being”. The word “existed” is misused, kindly correct to “existing” or “that are present in the”;

-        Line 41, the word: “anti-oxidation”. Would It be “antioxidant” a more appropriate term?

-        Line 42, the sentence: “As one of the most effective anticancer drugs in the world”. To the best of my knowledge, there are many classes of drugs that are used against cancer, and the appropriateness of a selected treatment is case-specific. Unless authors have a clear body of reference to support such claim, I kindly suggest this sentence to be omitted;

-        Line 47-48, the sentence: “To improve therapy efficiency and to reduce side effect, it is very important to frequently detect the concentration of 5-FU in blood samples from patients”. Could authors kindly add references to support this statement?

-        Line 48, the sentence: “Surface-enhanced Raman scattering (SERS) is a magical phenomenon”. I kindly suggest authors to avoid describing physical phenomena with this vocabulary;

-        Lines 66-67, the sentence: “The high enhancement efficiency makes the SERS to be the most sensitive analytical technique even for the detection of a single molecule”. I kindly ask authors for more references to support this claim. The cited reference is outdated, and more recent bibliographic support is needed;

-        Line 70, the sentence: “The most substrates are made”. I suggest changing to “most of the”;

-        Line 81, the sentence: “due to its low cost-effectiveness”. I suggest authors to rewrite to: “due to the affordability, effectiveness …”;

-        Line 88-89, the sentence “unstable of fluorescence”. I suggest authors to change to “instability of fluorescence”;

-        All Results and Discussion section. The results were presented and briefly described, however there was no thorough discussion of the findings. Very few references were used and most of the observations lack comparisons with the state of the art. Furthermore, even though authors compared their findings with a LC-MS assay, it would be very informative to provide further information such as comparing the analytical performance of the method described by the authors and that of other recent works in literature based on similar approaches.

Therefore, I suggest major revision.

Author Response

Responses to the comments and Suggestions for Authors 1

  • The sentence: “heterocyclic substances existed in DNA or RNA chains in human being” has been changed to “heterocyclic substances existing in DNA or RNA chains in human being”. (see revised version Line 38)

(2) The word: “anti-oxidation” has been changed to “antioxidant”. (see revised version Line 41)

(3) The sentence: “As one of the most effective anticancer drugs in the world,” has been omitted from the text, and “in the last fifth years” has been changed to “In the last fifth years”. (see revised version Line 42)

(4) I think it may be better to omit the sentence: “To improve therapy efficiency and to reduce side effect, it is very important to frequently detect the concentration of 5-FU in blood samples from patients”. Thus, in the revised manuscript, the sentence mentioned has been omitted. (see revised version Line 46)

(5) The sentence: “Surface-enhanced Raman scattering (SERS) is a magical phenomenon” has been changed to “Surface-enhanced Raman scattering (SERS) is a phenomenon” has been omitted. (see revised version Line 55)

(6) The sentence: “The high enhancement efficiency makes the SERS to be the most sensitive analytical technique even for the detection of a single molecule”. The original reference 14 has been omitted, and replaced by three new references (14-16). (see revised version Line 65)

Accordingly, the original references (15~34) have been changed to references (17~36).

(7) The sentence: “The most substrates are made” has been changed to “Most of the substrates are made”. (see revised version Line 67)

(8) The sentence: “due to its low cost-effectiveness” has been changed to “due to the affordability, effectiveness …”. (see revised version Line 78)

(9) The sentence “unstable of fluorescence” has been changed to “instability of fluorescence”. (see revised version Line 85-86)

(10) In the revised version, a comparison of different methods including our SERS-LFIA for 5-FU detection has been presented in Table 1. The corresponding literatures are given in the Table 1 and in References. (see revised version Line 385, Table 1).

Accordingly, the original Table 1-3 have been changed to Table 2-4.

Reviewer 2 Report

This work reports a Surface-Enhanced Raman Scattering (SERS)-based lateral flow assay for the rapid and sensitive detection of 5-Fluorouracil. The key of this assay is the incorporation of Au@Ag/Au nanoparticle into a lateral flow immunochromatographic assay. By optimizing the composition/size of immunoprobe and the volume of 5-FU-OVA/immunoprobe, a limit of detection of 4.4 pg/mL was achieved. In my opinion, the manuscript is well-written, the figures are easy to read and the results are interesting. The publication is recommended after addressing the following comments and questions.

1)     The authors mentioned the right dosage of 5-FU is critical to improving the therapy efficiency (line 44-51). Then what is the commonly accepted cut-off value of 5-FU? Does this value fall in the working range of this assay, i.e., 0-100 ng/mL?

2)     The authors claimed a SERS-LFIA for the detection of 5-FU in blood sample. This is true but a bit misleading because this assay requires extensive sample pretreatment and actually this assay was detecting 5-FU in PBS (line 208-215). Please revise the title. 

Author Response

Responses to the comments and Suggestions for Authors 2

  • The authors mentioned the right dosage of 5-FU is critical to improving the therapy efficiency (line 44-51). Then what is the commonly accepted cut-off value of 5-FU? Does this value fall in the working range of this assay, i.e., 0-100 ng/mL?

Reply: For this comment, a sentence “In cancer treatment, the effective dose of 5-FU in blood samples is about 150 ng mL−1 [27]. Apparently, the sensitivity of SERS-LFIA is sufficiently satisfied with the requirement for 5-FU detection.” was added into the revised manuscript. (see revised version Line 343-345)

  • The authors claimed a SERS-LFIA for the detection of 5-FU in blood sample. This is true but a bit misleading because this assay requires extensive sample pretreatment and actually this assay was detecting 5-FU in PBS (line 208-215). Please revise the title. 

Reply: For quantitative analysis, because of interference from other substances in the real samples, usually the target analyte should be extracted and enriched from real samples, and then be diluted in an appropriate solution and detected by an analytical method. Therefore, it may be not necessary to revised title.

Round 2

Reviewer 1 Report

The authors have addressed my comments. I have no further remarks.